# Loss of Wwox Causes Defective Development of Cerebral Cortex with Hypomyelination in a Rat Model of Lethal Dwarfism with Epilepsy

**DOI:** 10.3390/ijms20143596

**Published:** 2019-07-23

**Authors:** Yuki Tochigi, Yutaka Takamatsu, Jun Nakane, Rika Nakai, Kentaro Katayama, Hiroetsu Suzuki

**Affiliations:** Laboratory of Veterinary Physiology, School of Veterinary Medicine, Faculty of Veterinary Science, Nippon Veterinary and Life Science University, Musashino-shi, Tokyo 180-8602, Japan

**Keywords:** Wwox, cerebral cortex, neuron, oligodendrocyte

## Abstract

WW domain-containing oxidoreductase (Wwox) is a putative tumor suppressor. Several germline mutations of Wwox have been associated with infant neurological disorders characterized by epilepsy, growth retardation, and early death. Less is known, however, about the pathological link between Wwox mutations and these disorders or the physiological role of Wwox in brain development. In this study, we examined age-related expression and histological localization of Wwox in forebrains as well as the effects of loss of function mutations in the Wwox gene in the immature cortex of a rat model of lethal dwarfism with epilepsy (*lde/lde*). Immunostaining revealed that Wwox is expressed in neurons, astrocytes, and oligodendrocytes. *lde/lde* cortices were characterized by a reduction in neurite growth without a reduced number of neurons, severe reduction in myelination with a reduced number of mature oligodendrocytes, and a reduction in cell populations of astrocytes and microglia. These results indicate that Wwox is essential for normal development of neurons and glial cells in the cerebral cortex.

## 1. Introduction

The WW domain-containing oxidoreductase gene (*WWOX*) is a putative tumor suppressor gene, spanning the second most common chromosomal fragile site (FRA16D) in the human genome. Moreover, *WWOX* expression is frequently suppressed in many types of cancer cells [1,2]. Although several genetically modified mouse models have been generated to assess the in vivo tumor suppressor activity of Wwox protein [3,4,5,6,7,8], conditional knockout of *Wwox* in mammary epithelium found that Wwox did not behave as a classical tumor suppressor [9]. *Wwox* is considered physiologically indispensable for survival, as Wwox protein is widely expressed throughout the entire body [1,10,11] and the loss of Wwox protein in rodents frequently results in growth retardation and early postnatal death [5,8,12,13]. Because *Wwox* mutations also cause hypogonadism, Wwox protein is considered necessary for normal steroidogenesis in gonads [6,7,14,15]. Wwox protein has pleiotropic functions including gene transcription, protein stability, cell metabolism, survival, proliferation, apoptosis, and genomic stability to maintain cell homeostasis via association with multiple signaling pathways [16,17,18], suggesting that various type of mutations in the *Wwox* gene may be involved not only in tumorigenesis but in susceptibility to many diseases [19].

Wwox protein is also expressed in the central nervous system (CNS). In humans, for example, Wwox is expressed in the neurons and astrocytes of the cerebral frontal and occipital cortices, in several nuclei of the medulla, and in ependymal cells from a lateral cerebral ventricle [10]. In addition, Wwox protein is highly expressed in the developing nervous system of mouse embryos, with the level of expression and localization of this protein in mouse CNS drastically changing from late embryonic- to postnatal-stage [20]. These results suggest that *Wwox* may be important for the development of the CNS and for the maintenance of its normal function. The expression of Wwox was found to promote neuronal differentiation in vitro [21], and down regulation of Wwox was found to induce the hyperphosphorylation of Tau, leading to the formation of neurofibrillary tangles in the neurons of patients with Alzheimer’s disease (AD) [22]. Various types of *Wwox* gene mutations were recently found to be responsible for human pediatric neurodevelopmental disorders with a broad spectrum of clinical features, including growth retardation, microcephaly, epileptic seizures, ataxia, mental retardation, intellectual disability, retinopathy, and early death [23,24,25,26]. 

Our study using the inbred rat strain, lethal dwarfism with epilepsy (lde), was the first to show that Wwox was necessary for the development and function of the CNS [13]. Initially, *lde/lde* rats were identified as spontaneous mutants with severe dwarfism, gait ataxia, pediatric epilepsy, male hypogonadism, and early postnatal death [12,14]. Phenotypic analysis showed that epileptic seizures could be evoked by exposing *lde/lde* rats to sound after postnatal day (PND) 16, and that epilepsy in these rats was characterized initially by wild running and progressed to tonic–clonic convulsions [12,13]. Pathological analysis of *lde/lde* rat brains at PND 28 revealed many extracellular vacuoles in the CA1 region of the lateral hippocampus and amygdala [12,13]. These pleiotropic phenotypes were inherited as autosomal recessive traits. Linkage analysis and a candidate approach showed that a 13-bp deletion in exon 9 of the *Wwox* gene was responsible for *lde* phenotype [13]. Although expression of mutant *Wwox* mRNA was detected in the testes and hippocampus of *lde/lde* rats, no Wwox protein was detected by western blot analysis [13]. Subsequently, Wwox knockout mice showed neurological disorders with audiogenic epileptic seizures, in addition to severe dwarfism and early postnatal death [25]. These results strongly suggest that Wwox expression is required for the normal development and function of CNS and that mutations in *Wwox* result in neurological disorders in infants. Recently, several reports reviewed the phenotypic spectrum of *Wwox*-related epileptic encephalopathy (WOREE) syndrome [27,28], and brain magnetic resonance image (MRI) have indicated the presence of organic defects in the forebrain of patients with WOREE syndrome [27,28,29,30]. To date, however, no study has correlated histopathological alterations in forebrains with *Wwox* mutations. In the present study, we analyzed Wwox expression in normal forebrains, as well as pathological changes in the cerebral cortex of *lde/lde* rats. 

## 2. Results

### 2.1. Expression and Localization of Wwox Protein

To assess the expression of Wwox protein in PND 21 *+/+* male rats, protein extracts of various organs were subjected to western blot analysis. Wwox protein was detected in all organs examined, as reported previously [1,10,11], although the levels of expression varied (Figure 1A). Western blot analysis also showed that Wwox protein was expressed in different parts of the CNS, including the olfactory bulb, cerebral cortex, hippocampus, diencephalon, cerebellum, brain stem, and spinal cord (Figure 1B). Immunohistochemical analysis showed that Wwox protein was widely expressed in the forebrain (Figure 1C–I), especially in layers II-III and V of the cerebral cortex, as well as the white matter (Figure 1D), corpus callosum (CC) (Figure 1E), hilus in the hippocampus (Figure 1F), habenular nuclei (HN), thalamus (Figure 1G), hypothalamus (Figure 1H), and internal capsule (IC) (Figure 1I). In contrast, no protein band with the same electrophoretic mobility was detected in the whole brain and cerebral cortex of *lde/lde* rats, although a very weak band of slightly lower mobility was detected (Figure 2A,B asterisk). Assessments of brains of *+/lde* rats showed a single band of normal molecular weight, but its intensity was almost half that observed in *+/+* rats (Figure 2A,B). Immunohistochemistry showed intense signals of Wwox protein in the cerebral cortex layer V and corpus callosum of *+/+* and *+/lde* rats, but not of *lde/lde* rats (Figure 2C). Wwox protein was mainly localized to the cytoplasm (Figure 1 and Figure 2) as previously reported [5,6,10,31]. Additionally, we found that the expression of Wwox protein increased with age in whole brains and cerebral cortices of *+/+* rats (Figure 2D,E). These findings suggest that Wwox protein might have important roles in normal brain development during early postnatal period.

### 2.2. Loss of Wwox Protein Affects Postnatal Development of the Cerebral Cortex

To assess the pathological alterations associated with Wwox mutations in the cerebral cortices of *lde/lde* rats, we examined the distributions of neurons and glial cells (astrocytes, oligodendrocytes, and microglia) in the cerebral cortex during early development.

Immunohistochemistry using an antibody against a neuronal specific nuclear protein, NeuN [32] did not show any significant difference of neuron number between *+/+* and *lde/lde* rats (Figure 3A,D), and these results were also confirmed by western blot analysis (Figure 3H,J). In consistent with these findings, there is no significant difference between *+/+* and *lde/lde* rats in the thickness of cerebral cortices at all ages (Figure 3F) and in neuron number in each cortical layer at PND21 (Figure 3G). In contrast, western blotting analysis for neurite marker MAP2 mainly detected in immature dendrites [33,34] revealed that the level of its expression was significantly lower in cerebral cortices of *lde/lde* than of *+/+* rats at all ages (Figure 3H,I). Immunostaining for MAP2 also showed significantly reduced signals in cortical layers I to II/III of *lde/lde* than of *+/+* at all days examined (Figure 3B,E). These results were also confirmed by immunostaining for FluoroPan Neuronal Marker indicating that axon-like vertical neurite growth is present but totally reduced in *lde/lde* at PND 21 (Figure 3C). Taken together, it is suggested that loss of Wwox impairs neural differentiation but not the number. 

The degree of myelination in rat brains was assessed using antibodies to the specific myelination markers, MBP and CNP [35,36]. A few dispersed signals corresponding to both were detected near the white matter (WM) of *+/+* rats on PND 5, with positivity gradually extending toward the surface of the cerebral cortex. In contrast, *lde/lde* rats showed significantly lower myelination on PNDs 5–21 (Figure 4A–D). Western blotting also showed age-associated increases in MBP and CNP in cerebral cortices of *+/+* rats, along with significantly lower levels of expression in *lde/lde* rats (Figure 4E–G). APC is an oligodendrocyte marker that can be used to visualize mature oligodendrocyte [37,38]. Immunohistochemistry showed that the number of APC-positive cells was significantly lower in a subarea of the cerebral cortex including the WM region of *lde/lde* than of *+/+* rats on PNDs 15–21 (Figure 4H,I). The reduced number of APC-positive cells was also observed in the region of corpus callosum. These results indicate that the number of mature oligodendrocytes is lower in *lde/lde* than in *+/+* rats and myelination is impaired in *lde/lde* rats.

Rat brain tissues were also incubated with antibodies to the astrocyte marker GFAP [39] and the microglia marker Iba1 [40]. The GFAP-positive cell number and its expression level in cerebral cortices gradually increased with age in both *+/+* and *lde/lde* rats, whereas the number and the expression was significantly lower in *lde/lde* than in *+/+* rats on PNDs 5 to 21 and PND 15, respectively (Figure 5A,B,E). Similarly, the expression level of Iba1 in cerebral cortices were significantly lower in *lde/lde* than in *+/+* rats at the all ages examined (Figure 5E), as was the number of Iba1-positive cells in cerebral cortices on PND10 to 21 (Figure 5C,D).

### 2.3. Cellular Localization of Wwox Protein in Cerebral Cortex and CC

Assessment of the cellular localization of Wwox protein in *+/+* forebrains on PND 21 showed a high degree of colocalization of NeuN and Wwox in the cytoplasm of pyramidal neurons located at layers II-III and V of the cerebral cortex. Single-dot or granular Wwox-signals were also detected in their nucleus (Figure 6A). Although the ratio of double positive cells to NeuN-single positive cells varied, similar co-localization was observed in other regions throughout forebrain. Wwox protein was also detected in the cytoplasm of most APC-positive oligodendrocytes in the CC (Figure 6B) and in other regions throughout the forebrain. Wwox protein showed a dot-like condensed localization in the cytoplasm of GFAP-positive astrocytes located in layer V of the cerebral cortex (Figure 6C). This distinct cytoplasmic localization of Wwox in GFAP-positive astrocytes was observed in both gray and white matter. In contrast, Iba1-positive microglia in the cerebral cortex and CC were not immunostained with antibody to Wwox (Figure 6D). Because neurons, oligodendrocytes, and astrocytes all derive from neural stem cells, these findings suggest that Wwox may be expressed in all neural stem cell-lineages.

## 3. Discussion

Consistent with previous reports, showing the systemic expression of Wwox [1,5,10], this study found that Wwox was expressed in all organs examined. Especially high levels of Wwox were observed in the brain, liver, small intestine, adrenal glands, kidneys, and testes, suggesting that Wwox has pleiotropic functions, involved in the neuroendocrine system and in metabolism [11,19].

Wwox protein was shown to localize in the CNS of adult humans [10] and of embryonic and adult mice [20]. Our data clearly showed age-related increases in Wwox expression in the normal rat brain during early postnatal development, indicating that Wwox may play an important role for establishing normal brain networks. Immunostaining detected Wwox protein in the soma of neurons located in the cerebral cortex, hippocampus, thalamus, and hypothalamus, as well as in brain regions containing abundant myelin sheath, such as the CC, white matter, and internal capsule at PND 21, results consistent with previous findings [10,20], suggesting that Wwox is involved in the functions of these neurons and their conducting pathways. Although previous immunohistochemistry found that Wwox protein was present in neurons and astrocytes but not in oligodendrocytes [10], our double immunofluorescence using specific markers clearly showed that Wwox protein was present in oligodendrocytes, as well as in neurons and astrocytes. To our knowledge, this study is the first to show that Wwox is expressed in oligodendrocytes.

In agreement with previous WB data [13], this study found that Wwox protein was highly expressed in the brains of *+/+* and *+/lde* rats, but not in the brains of *lde/lde* rats. The latter, however, showed weak expression of a slightly heavier protein (46.2 kDa) (Figure 2A,B), which may have been due to greater sensitivity of the new antibody, directed against amino acids 32–110 of human Wwox protein, a sequence 100% identical to that of rat Wwox. The Wwox gene of *lde/lde* rats contains a 13-bp deletion (c.1190_1202del) in exon 9 causes frame-shift, resulting in an aberrant C-terminal amino acid sequence (p.leu371Thrfs*53), theoretically 0.8 kDa larger than wild type, suggesting that this faint signal may be that of a mutated protein escaping from ubiquitin-mediated protein degradation due to its instability [13]. This possibility is also supported by the slight immunofluorescence of *lde/lde* brain sections when incubated with antibody to Wwox (Figure 2C). Although a variety of germline Wwox mutations have been recently identified in human, the most severe clinical presentation such as WOREE syndrome seems to be associated with genotypes consisting of early premature stop codons corresponding to Wwox knockdown [28]. Two C-terminal mutations causing frame-shift (c.1094_1095del, p.Val365Alafs*47 and 1138dup, p.Cys380Leufs*149) have been also identified in patients with WOREE syndrome [28], suggesting resultant mutated proteins are not functional even though they would be expressed. Therefore, it is unlikely that this faint expression of mutated Wwox protein would have substantial effects on the phenotype of *lde/lde* rats. Considering genetic and phenotypic similarities of *lde/lde* rats with WOREE syndrome, we conclude that *lde/lde* rat is a model for WOREE syndrome.

Using antibodies specific for neurons and each type of glial cell, the present study showed that Wwox mutation resulted in pathological alterations in the cerebral cortex. Although Wwox localized to the cell bodies of neurons in the cerebral cortices of *+/+* rats, the normal density and distribution of NeuN-positive neurons and the normal level of expression of NeuN in the cerebral cortices of *lde/lde* rats during early postnatal period indicated that Wwox is not required for proliferation and migration of immature neurons. In contrast, the level of MAP2 expression was significantly decreased in *lde/lde* cerebral cortices. Wwox has been reported to interact directly with GSK3β and regulates its phosphorylation of Tau. Hyperphosphorylated Tau has a decreased affinity for microtubules and disrupts microtubule stability, inhibiting neurite outgrowth and neuronal cell differentiation in SH-SY5Y cells [21,22]. The high level of Wwox protein in pyramidal neurons in *+/+* cerebral cortices and the reduction in MAP2-positive neurites in layers I to II–III of *lde/lde* cerebral cortices suggest that Wwox may be required for neuronal differentiation, including GSK3β-associated neurite growth. Because Wwox protein expression was also observed in cell bodies of the *+/+* hippocampus, thalamus, and hypothalamus, the behavioral and endocrinological defects in *lde/lde* rats may be involved in neural network defects in these regions.

In addition to the reduced expression of MAP2, the severe reduction in expression of the major myelination markers, MBP and CNP, indicated that developmental defects in *lde/lde* rats involve not only neural dendrites but the myelination of oligodendrocytes. In general, mature oligodendrocytes form myelin sheaths, which insulate neuronal axons, enabling saltatory conduction and providing metabolic support to axons to maintain their integrity [41]. The interaction between axons and oligodendrocytes is important for myelination [42,43], suggesting that the severe reduction in myelination of *lde/lde* cortices may result, at least in part, from the retarded growth of axons predicted by the delayed differentiation of neurons, as indicated by the reduced expression of MAP2 and reduced immunostaining of FluoroPan Neuronal marker. In addition, the reduced number of APC-positive oligodendrocytes in *lde/lde* cortices indicates that the marked reduction in myelination is also caused by the reduced number of mature oligodendrocytes. In rats, the active proliferation of oligodendrocyte precursor cells is completed around PND 20, whereas myelination starts around PND 10 and reaches a peak at PND 20, followed by low levels of myelination in adults [44]. These findings are generally consistent with our present data, showing a postnatal increase in APC-positive oligodendrocytes and the myelination markers MBP and CNP in *+/+* cerebral cortices. In *lde/lde* rats, epileptic seizures and ataxic gait occur after PND 16 [12,13]. Such neurological defects have been reported in several mutant animals showing hypomyelination [45,46,47,48]. Therefore, part of the phenotype of *lde/lde* rats is associated with hypomyelination, accompanied by a reduced number of mature oligodendrocytes.

This study also suggests that the loss of Wwox results in the significant reductions in GFAP-positive astrocytes and Iba1-positive microglia. This was confirmed by both western blotting and immunohistochemistry. Astrocytes have been classified as protoplasmic and fibrous types, present in gray and white matter, respectively [43]. In the present study, Wwox was found to be expressed in GFAP-positive astrocytes of both the cortex (Figure 6C) and white matter. The condensed dot-like distribution of Wwox in astrocytes differed from the cytoplasmic diffuse distribution of Wwox in neurons and oligodendrocytes. Wwox has been detected in the nucleus, mitochondria, and Golgi apparatus, as well as in cytoplasm [49,50]. Its unique localization in astrocytes suggests that Wwox may localize to certain cellular structure. The lack of Wwox expression in *+/+* microglia suggests that the decrease of microglia in *lde/lde* rats is a secondary phenomenon. In contrast to microglia derived from the outside of developing brain, the expression of Wwox in neurons, astrocytes, and oligodendrocytes suggests that this protein is specifically expressed in neural stem cell-lineages and may play a role in their development directly. In addition to the autonomous differentiation of neurons and glial cells, their crosstalk is important for their differentiation [42,43,51]. Therefore, reductions in the numbers of astrocytes and microglia may influence or be influenced by the reduced differentiation of neurons and oligodendrocytes. Recently, Hussain et al. have reported reduced number of GABA-ergic inhibitory interneuron and activation of microglia and astrocytes in the hippocampus of two-week-old Wwox knockout mice, suggesting pathological involvement of gliosis and neuroinflammation [52]. These results are apparently in contrast to our present data showing significant reductions in dendrite growth of wide-ranging neurons and GFAP- and Iba1-positive areas in the cerebral cortex, even though different regions in forebrain examined. To reveal this discrepancy, more detail analyses in various brain region including the hippocampus and cerebral cortex needs to be done in both rodent models. 

Although no pathological feature to date has been shown in human congenital diseases with Wwox mutations, MRI scanning of patients with these disorders have shown hypoplasia, dysplasia, and atrophy of many brain regions [23,24,26,27,28,29,30]. Interestingly, Piard et al. has reported that corpus callosum hypoplasia and progressive cerebral atrophy are most prominent anomalies [28], and delayed myelination and progressive demyelination have been also reported [26,27,28,29,30,53]. These findings suggest that retarded neurite growth and hypomyelination may be involved in brain anomalies occurring in patients with Wwox mutations. In addition, the diversity in the MRI observation might be caused by severity and range of these defects, possibly related with individual differences in type of mutation, genetic background, age of onset, and so on.

## 4. Materials and Methods

### 4.1. Animals

Male rats were derived from the inbred LDE strain [12,13,14] and maintained in a clean conventional animal room under a controlled light-dark cycle (14:10 h), with a certified diet (CR-LPF; Oriental Yeast Co. Ltd., Tokyo, Japan) and water ad libitum [54]. The rats were genotyped by PCR-based methods to detect mutations in the Wwox gene, as described [13]. Because *lde/lde* rats have high mortality rates [12,13,14], the numbers of littermates were reduced to <5 pups at around PND 5 to enhance the survive of *lde/lde* pups. At least three affected and three normal males were included in each experiment (western blot and immunostaining) and examined each day. All animal experiments and animal care were approved by Animal Care and Use Committee of Nippon Veterinary and Life Science University (Protocol #29K-44, 31 March 2018), and were conducted in accordance with the Guidelines of the Animal Care and Use Committee of Nippon Veterinary and Life Science University.

### 4.2. Reagents and Antibodies

All reagents for immunohistochemistry (IH) and western blotting (WB) were of the highest available grade. Antibodies and dilutions included rabbit anti-Wwox (1:1000 for IH and WB, HPA050992, Sigma-Aldrich, St. Louis, MO, USA), mouse anti-neuronal nuclei (NeuN) (1:2000 for IH, clone A60, Millipore, Billerica, MA, USA), rabbit anti-NeuN (1:2000 for WB, clone EPR1263, Abcam plc, Cambridge, UK), rabbit anti-microtubule associated protein 2 (MAP2) (1:1000 for IH and WB, Abcam), FluoroPan Neuronal Marker (1:50 for IH, Millipore), mouse anti-myelin basic protein (MBP) (1:50 for IH and WB, clone 1, Millipore), mouse anti-2′,3′-cyclic-nucleotide 3′-phosphodiesterase (CNP) (1:500 for IH and WB, clone 11-5B, Sigma-Aldrich), mouse anti-adenomatous polyposis coli (APC) (1:1000 for IH, clone CC-1, Millipore), goat anti-glial fibrillary acidic protein (GFAP) (1:2000 for IH, 1:1000 for WB, Abcam), goat anti-ionized calcium-binding adapter molecule 1 (Iba1) (1:2000 for IH, Abcam), rabbit anti-Iba1 (1:1,000 for WB, Wako Pure Chemical Industries, Ltd., Tokyo, Japan), and mouse anti-β-actin (1:5000 for WB, Applied Biological Materials Inc., Richmond, BC, Canada).

### 4.3. Immunohistochemistry

Normal (*+/+*, *+/lde*) and *lde/lde* rats at PNDs 5, 10, 15, and 21 were anesthetized with isoflurane, under the control of a small animal anesthetizer (TK-7, Biomachinery, Chiba, Japan). The caudal vena cava was cut for bleeding, followed immediately by manual perfusion of phosphate-buffered saline (PBS) containing 10 unit/mL heparin from the left ventricle. After almost all blood was replaced by the PBS and the animals died, they were perfused with freshly prepared 4% buffered paraformaldehyde. Their brains were removed, fixed in 4% paraformaldehyde for 48 h, and embedded in paraffin, followed by sectioning into 5-µm-thick sequential coronal sections spanning the whole length of the hippocampus [13]. For antigen retrieval, deparaffinized sections were incubated with 10 mM sodium citrate (pH 6.0) at 121 °C for 5 min. Following three washes with PBS, tissue sections were blocked with 10% donkey serum (Sigma-Aldrich) in PBS for 30 min and incubated overnight at 4 °C with primary antibodies. After washing with PBS, the sections were incubated with appropriate secondary antibodies [55,56] for 1 h at room temperature. The resultant sections were mounted in ProLong Gold Antifade Reagent containing DAPI (Life Technologies, Carlsbad, CA, USA) for counterstaining. 

Images were collected on a Biozero BZ-X710 fluorescence microscope using software BZ-X analyzer Ver. 1.3.1.1 (Keyence, Tokyo, Japan). The positive areas stained with antibodies against MAP2, MBP, and CNP were quantified as pixels per area (mm^2^) using analytical software Image J (Ver. 1.46r, NIH, Bethesda, MD, USA). NeuN-, APC-, GFAP-, and Iba1-positive cell number was also counted using Image J in same size of brain area shown as representative pictures in each figure. To quantify the distribution of neuron, cerebral cortices were divided into 5 layers (I, II/III, IV, V, VI) based on specific morphology of NeuN-positive cells. The area (mm^2^) of each layer and the number of NeuN-positive cells were measured by Image J as described above. In all images used for analyzing, the background was reduced equally and then entire retained signals were measured. All quantitative analysis for immunohistochemistry was performed for the images taken from identical region among the brain sections prepared from at least three rats.

### 4.4. Western Blot Analysis

Brains and other organs obtained from rats sacrificed on PNDs 5, 10, 15, and 21 rats were frozen immediately in liquid nitrogen and stored at −80 °C. To prepare tissue extracts, organs were minced and sonicated in RIPA lysis buffer (50 mM Tris-HCl pH 7.6, 150 mM NaCl, 1 mM EDTA, 1% sodium deoxycholate, 0.1% Triton X-100, and 0.1% sodium dodecyl sulfate (SDS)), supplemented with protease inhibitor cocktail (Roche, Diagnostics, Mannheim, Germany), containing 1 mM phenylmethylsulfonyl fluoride (PMSF), 0.2 mM sodium orthovanadate, and 100 mM sodium fluoride. Protein concentrations were determined by BCA protein assay kits (Thermo Scientific, Rockford, IL, USA). Proteins in the extracts were separated by SDS-PAGE and transferred to PVDF membranes (Hybond-LFP, GE Healthcare Life Science, Pittsburgh, PA, USA) using semi-dry blotting system (ATTO, Tokyo, Japan) at 25 V for 15 min as described [13,55]. After blocking with 1% non-fat milk in PBS containing 0.1% Tween 20 for 1 h, the membranes were incubated overnight at 4 °C with appropriately diluted primary antibody, as above, followed by incubation with appropriate secondary antibodies and visualization with an Odyssey fluorescent imaging system (version 1.2; LI-COR Biotechnology, Lincoln, NE, USA) [55]. Densitometric analyses on western blot were performed by software Image J as previously described [55].

## Figures and Tables

**Figure 1 ijms-20-03596-f001:**
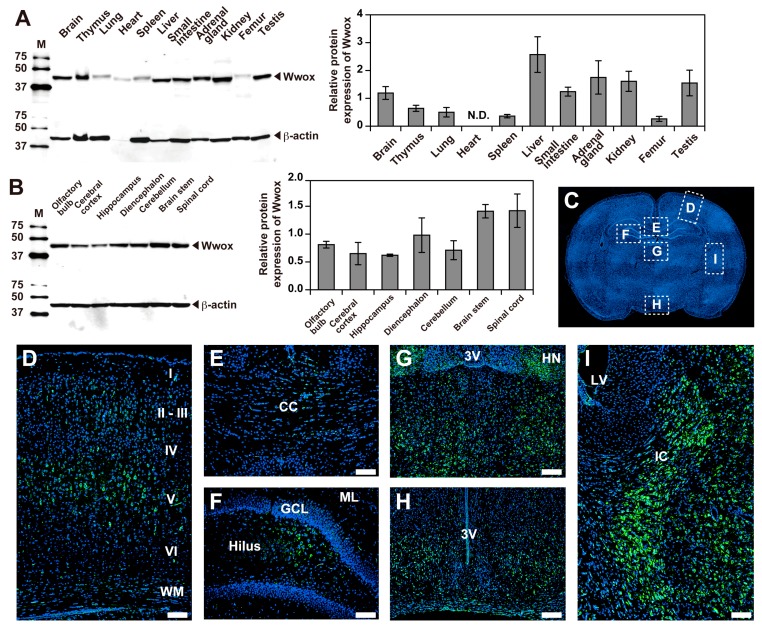
Expression of Wwox protein in major organs and localization of Wwox protein in brain at PND 21. (**A**,**B**) Western blotting analysis of Wwox protein expression in 11 organs (**A**) and seven portions of the brain (**B**) obtained from PND 21 *+/+* male rats; β-actin was used as a loading control. Representative pictures from three independent experiments are shown in both western blotting. The intensity of each band was quantified and normalized by β-actin. The graphs were shown as mean ± S.D. from three independent experiments. N.D., Relative expression of Wwox was not determined in heart because of apparently low expression of β-actin in this organ. (**C**–**I**) Localization of Wwox protein in areas of the rat forebrain (**C**), including the cerebral cortex (**D**), corpus callosum (**E**), hippocampus (**F**), thalamus (**G**), hypothalamus (**H**), and internal capsule (**I**). Green color indicates positivity for Wwox. Nuclei were stained with DAPI (blue). Representative pictures from three independent experiments are shown in both western blotting and immunohistochemistry. Scale bars, 100 µm. WM, white matter; CC, corpus callosum; ML, molecular layer; GCL, granule cell layer; HN, habecular nuclei; 3V, third ventricle; LV, lateral ventricle; IC, internal capsule.

**Figure 2 ijms-20-03596-f002:**
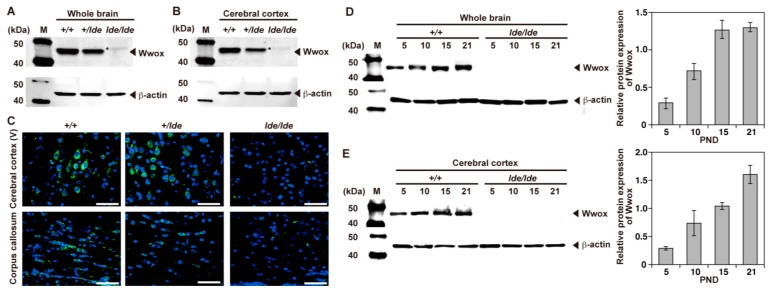
Deficiency of Wwox protein in *lde/lde* brain. (**A**,**B**) Western blotting analysis of Wwox protein expression in whole brains and cerebral cortices obtained from PND 21 male *+/+*, *+/lde*, and *lde/lde* rats. The asterisk indicates an extremely weak band with slightly higher molecular weight recognized by the Wwox specific polyclonal antibody. β-actin was used as a loading control. (**C**) Immunohistochemical images of layer V in the cerebral cortex (upper panels) and corpus callosum (CC) (lower panels) for each genotype. Green color indicates positivity for Wwox. Nuclei were stained with DAPI (blue). Scale bars, 50 µm. (**D**,**E**) Western blotting analysis showing the expression of Wwox in the whole brain (**D**) and cerebral cortex (**E**) at PNDs 5, 10, 15, and 21. Representative pictures from three independent experiments are shown in both western blotting and immunohistochemistry. The intensity of each band was quantified and normalized by β-actin. The graphs were shown as mean ± S.D. from three independent experiments.

**Figure 3 ijms-20-03596-f003:**
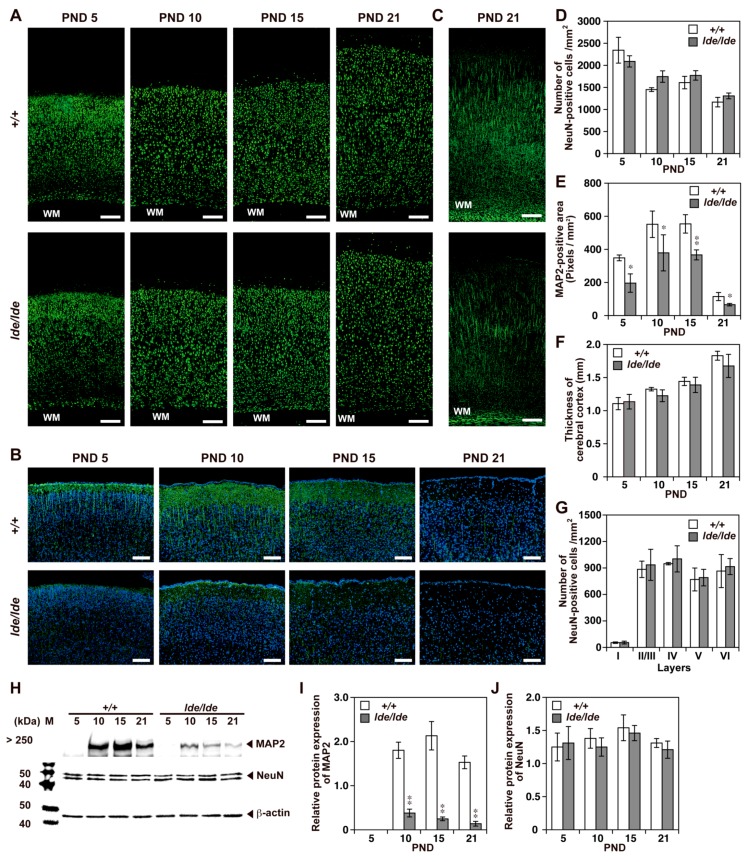
Density and distribution of neurons and elongation of neurites in the cerebral cortex at PNDs 5, 10, 15, and 21. (**A**–**C**) Immunohistochemistry using FluoroPan Neuronal marker (**C**) and antibodies to NeuN (**A**) and MAP2 (**B**). Representative images of the entire cerebral cortex (**A**,**C**) and cortical layers I to II-III (**B**) in +/+ and *lde/lde* rats are shown. Green color indicates positive signals. Nuclei were stained with DAPI (blue) (**B**). Scale bars, 200 µm (**A**,**C**), 100 µm (**B**). WM, white matter. (**D**,**E**) The NeuN-positive cell number (**D**) and the MAP2-positive area (**E**) were quantified and compared between *+/+* and *lde/lde* at each PND indicated. (**F**) Thickness of the cerebral cortex was measured on the section and compared between *+/+* and *lde/lde*. (**G**) Quantification of NeuN-positive cell number in the cortical layers I, II/III, IV, and VI at PND 21. (**H**–**J**) Western blotting analysis showing the expression of MAP2 (**H**,**I**) and NeuN (**H**,**J**) in the cerebral cortex at each PND indicated. β-actin was used as a loading control. The intensity of each band was quantified and normalized by β-actin. All graphs were shown as mean ± S.D. from three independent experiments. A statistical analysis was performed between *+/+* and *lde/lde* at each PND, respectively. Student’s *t*-test was used to determine statistical significance (* *p* < 0.05, ** *p* < 0.01).

**Figure 4 ijms-20-03596-f004:**
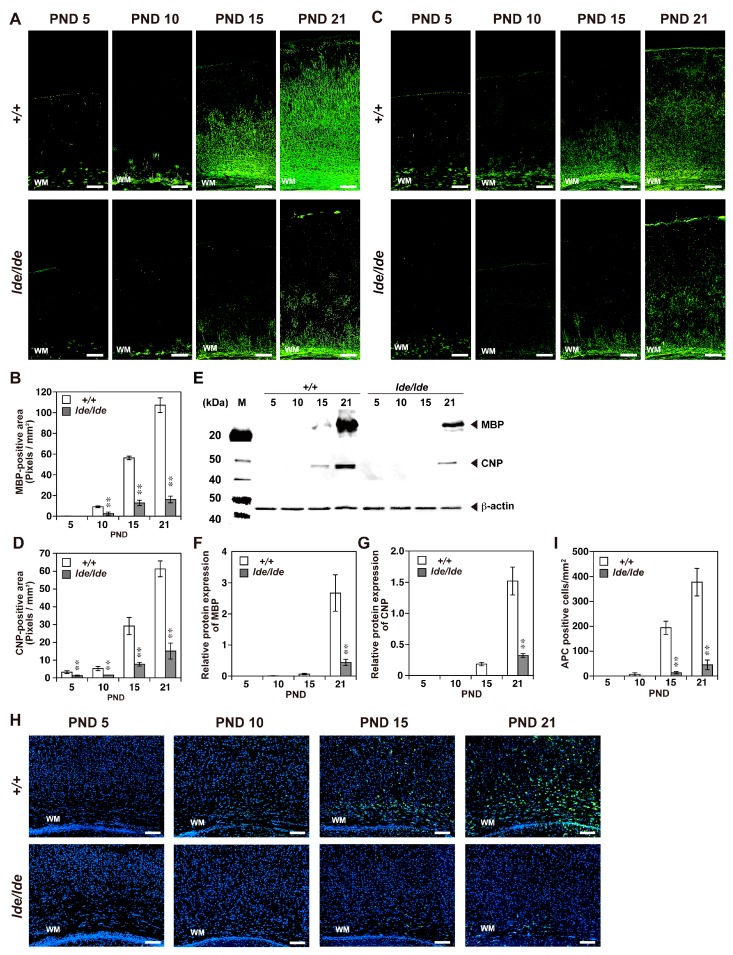
Age-related changes in myelination and number of oligodendrocytes in the cerebral cortex during early postnatal development. (**A**,**C**,**H**) Immunohistochemistry using antibodies to MBP (**A**), CNP (**C**), and APC (**H**) at PNDs 5, 10, 15, and 21. Representative images of *+/+* and *lde/lde* rats were shown in the upper and lower panels, respectively. Green color indicates positivity signals. Nuclei were stained with DAPI (blue) (**H**). Scale bars, 200 µm (**A** and **C**), 100 µm (**H**). (**B**,**D**) The MBP (**B**) and CNP (**D**) positive areas were quantified and compared between *+/+* and *lde/lde* rats. (**E**) Representative image of western blotting analysis showing the expression of MBP and CNP in the cerebral cortex. β-actin was used as a loading control. (**F**,**G**) The intensity of each band was quantified and normalized by β-actin. (**H**,**I**) The number of APC-positive cell was quantified in a subarea of the cerebral cortex containing WM (white matter) as shown in H. All graphs were shown as mean ± S.D. from three independent experiments. A statistical analysis was performed between *+/+* and *lde/lde* at each PND indicated, respectively. Student’s *t*-test was used to determine statistical significance (* *p* < 0.05, ** *p* < 0.01).

**Figure 5 ijms-20-03596-f005:**
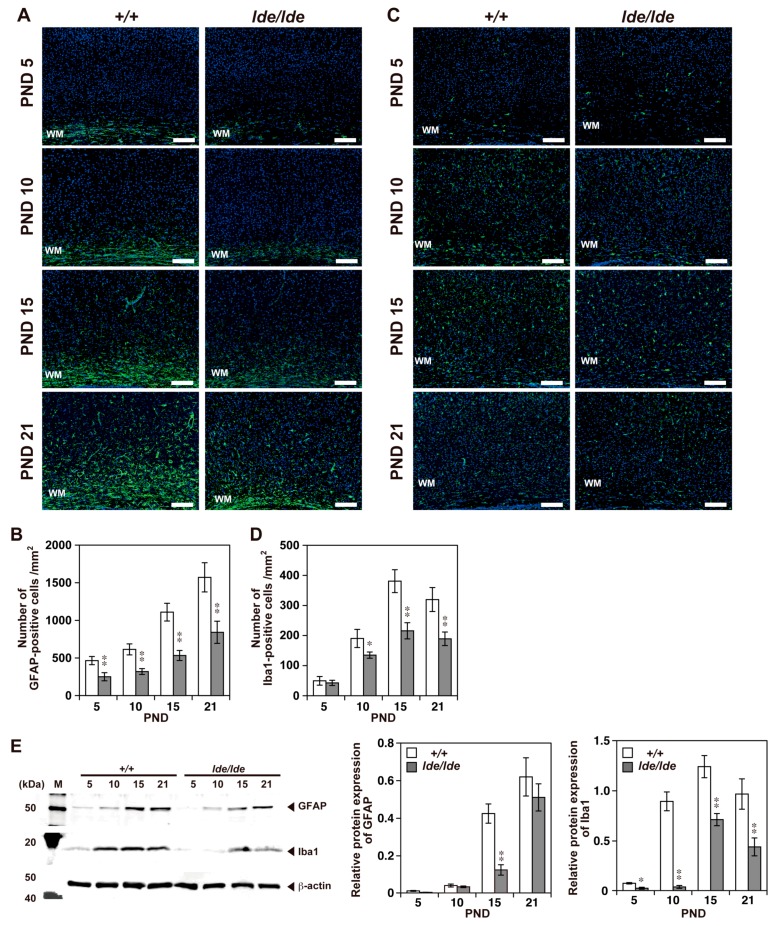
Distribution of astrocytes and microglia in the cerebral cortex during early postnatal development. (**A**,**C**) Immunohistochemistry using antibodies to GFAP (**A**) and Iba1 (**C**) at PNDs 5, 10, 15, and 21. Representative pictures from three independent experiments are shown. Green color indicates positive signals. Nuclei were stained with DAPI (blue). Scale bars, 100 µm. WM, white matter. (**B**,**D**) The numbers of GFAP- and Iba1- positive cells were quantified in a subarea of the cerebral cortex containing WM (white matter) as shown in **A** and **C** and compared between *+/+* and *lde/lde* rats at each PND indicated. (**E**) Representative image of western blotting analysis showing the expression of GFAP and Iba1 in the cerebral cortex at PNDs 5, 10, 15, and 21. β-actin was used as a loading control. The intensity of each band was quantified and normalized by β-actin. All graphs were shown as mean ± S.D. from three independent experiment. A statistical analysis was performed between *+/+* and *lde/lde* at each PND indicated. Student’s *t*-test was used to determine statistical significance (* *p* < 0.05, ** *p* < 0.01).

**Figure 6 ijms-20-03596-f006:**
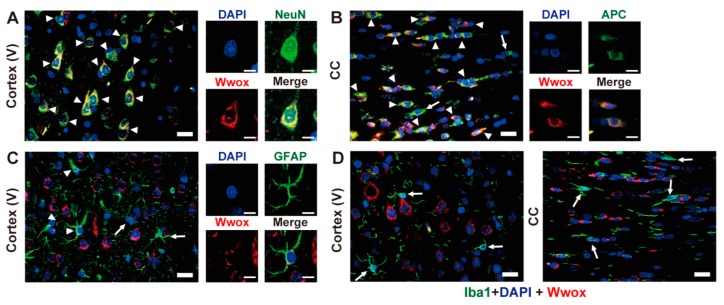
Localization of Wwox protein in neurons and glial cells of the cerebral cortex and CC. Tissue sections prepared from PND 21 *+/+* male rats were subjected to double immunostaining using antibodies against indicated markers (green) and Wwox (red). Nuclei were stained with DAPI (blue). Representative results of layer V in the cerebral cortex (**A**,**C**,**D**) and CC (**B**,**D**). Arrowheads and arrows indicate double positive and single positive cells, respectively. Scale bars, 20 and 10 µm in low and high magnification. Representative pictures from three independent experiments are shown. CC, corpus callosum.

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
