# Peer review of "Loss of Wwox Causes Defective Development of Cerebral Cortex with Hypomyelination in a Rat Model of Lethal Dwarfism with Epilepsy"

_ijms, 2019, doi:10.3390/ijms20143596_

Round 1
Reviewer 1 Report
I think this is a very nice study further defining the effects of Wwox deficiency on neural development. I recommend that it should be accepted with minor revisions to address the following points:
Expression of the loading control is so varied in Fig 1A that it is difficult to interpret relative levels of Wwox in the different tissues. I think that readers will understand the data better if you also provide a table with relative Wwox expression normalized to Beta-actin on the y axis and the different tissue types along the y axis.
There appears to be nuclear Wwox staining in the pyramidal neurons shown in Figure 6A- not only in the cell shown in the inset but many of the cells staining positive for Wwox, however the text does not address this. There are novel functions being described for Wwox in DDR that involve its nuclear localization (Schrock et al, Oncogene, 2017). The localization of Wwox needs to be clearly stated and framed into context with the current literature.
Altogether, nice work that contributes greatly to our understanding of Wwox function.
Author Response
Response to the comments from reviewer 1 on manuscript ijms-541172
The authors thank Reviewer 1 for careful reading and thoughtful comments. Our responses to Reviewer 1 are described below.
Comment #1: I think this is a very nice study further defining the effects of Wwox deficiency on neural development. I recommend that it should be accepted with minor revisions to address the following points:
Response to comment #1: Thank you for reviewing our manuscript, and I am glad to read your positive comment encouraging us. We are happy if our data is published in IJMS and might help diagnosis and improve therapeutic strategy of WOREE syndrome.
Comment #2: Expression of the loading control is so varied in Fig 1A that it is difficult to interpret relative levels of Wwox in the different tissues. I think that readers will understand the data better if you also provide a table with relative Wwox expression normalized to Beta-actin on the y axis and the different tissue types along the y axis.
Response to comment #2: Thank you for proper suggestion for improving the presentation of data in Figure 1. We calculated the relative expression levels of Wwox protein in different tissues and inserted those data as bar-graph in revised manuscript (MS) (Figure 1A and B), indicating high expression level of Wwox in brain, liver, small intestine, adrenal gland, kidney, and testis (Line 222-225 in revised MS). Since cardiac expression of beta-actin was very low, we could not determine cardiac relative expression level of Wwox (N.D.; not determined, see Figure 1 and the legend of revised MS).
Comment #3: There appears to be nuclear Wwox staining in the pyramidal neurons shown in Figure 6A- not only in the cell shown in the inset but many of the cells staining positive for Wwox, however the text does not address this. There are novel functions being described for Wwox in DDR that involve its nuclear localization (Schrock et al, Oncogene, 2017). The localization of Wwox needs to be clearly stated and framed into context with the current literature.
Response to comment #3: According to reviewer’s suggestion, we mentioned the nuclear localization of Wwox in pyramidal neurons (Line 201-202; Single-dot or granular Wwox-signals were also detected in their nucleus (Fig. 6A)). In addition, in order to describe the cellular function of Wwox in the introduction as suggested another reviewer, we added two papers in the references of revised MS (17. Abu-Remaileh et al., JBC 2015; 18. Schrock et al., Oncogene, 2017).
Other:We deleted several sentences in the Materials and Methods (Line 357-361). That had been remained during revision process in initial version of MS.
Reviewer 2 Report
This manuscript describes the expression and localization of WWOX, a WW-domain-containing oxidoreductase, in postnatal rat brain and starts to examine the consequences of homozygous Wwox loss of function mutation on brain structure.
The manuscript brings to attention an additional rodent model of Wwox deficiency (the Ide/Iderats) and suggests that this model is relevant for the understanding of the mechanisms contributing to the pathology of the WOREE syndrome. The value of this report would increase substantially if the authors could provide evidence that at least some of the mutations associated with human disease cause WWOX deficiency or at least discuss how these mutations might affect protein stability based on structural modeling data . In addition, the author’s attention is drawn to the following points:
1. Fig 2 and discussion lines 225-237: The authors suggest that the 13bp deletion in exon 9 leads to the translation of a larger and unstable protein product. If the slower migrating band in Ide/Idebrain lysates is due to the expression of a larger mutant protein that would differ from wt by ~1kDa, this band should also be visible in the Ide/+ lysates where currently it might be obscured by the faster migrating band. The authors should attempt to resolve this using gradient SDS-PAGE gels and should quantify the relative amounts of wt and mutant WWOX in all samples. The readers would also benefit from seeing the lower part of the gel to determine if any cleavage products, indicative of protein instability, can be detected in lysates obtained from Ide/+and Ide/Iderats. The should also attempt to convincingly demonstrate the mutant protein instability. This can easily be done for ex. By incubating dissociated liver cells with and without proteasome and lysosome inhibitors for increasing amounts of time and following the accumulation of the slower migrating WWOX band.
2. Fig. 3: The fluorescence data on the figure suggest that, although the toatal number of NeuN+ neurons is comparable, there might be a difference in layer organization in Ide/Ide rats compared to wt and that by PND21 there is a difference in cortical thickness that might be related to changes in layer V. Quantification of NeuN+ cells in each layer and of cortical thickness is necessary.
3. Fig. 5: Iba 1 staining failed and should be optimized and re-evaluated before drawing conclusions. The authors need to provide quantification of GFAP and Iba1+ cell densities, not just a quantification of the fluorescent area.
Minor points:
The introduction should provide more information regarding the biological functions of WWOX (e.g. role in signaling, functions of nuclear WWOX, etc).
Results, lines 91-92: The data presented in this section are insufficient to speculate that Wwox protein is important for postnatal CNS development. This conclusion can only be reached later.
Fig. 1 A: change “Adrenal grand” to “Adrenal gland”.
Fig. 1 D: the label for cortical layer I should be moved up.
Line 124: repace “cortexes” with “cortices”
Line 151: “around the WM” is vague. If corpus callosum was evaluated, please state so. If a more extended area was quantified, it should be boxed on the figure.
Author Response
Response to the comments from reviewer 2 on manuscript ijms-541172
The authors thank Reviewer 2 for careful reading and thoughtful comments. Our responses to Reviewer 2 are described below.
Comment #1: This manuscript describes the expression and localization of WWOX, a WW-domain-containing oxidoreductase, in postnatal rat brain and starts to examine the consequences of homozygous Wwox loss of function mutation on brain structure.
Response to comment #1: Thank you for reviewing our manuscript and providing many important comments to improve scientific quality of our manuscript (MS). We revised our MS accordingly to your comments as much as possible. We aspire for the revised MS will be published soon, because the data included in the MS might improve diagnosis and therapy for WOREE syndrome.We hope you will find our revised paper is now suitable for publication in IJMS.
Comment #2: The manuscript brings to attention an additional rodent model of Wwox deficiency (the Ide/Iderats) and suggests that this model is relevant for the understanding of the mechanisms contributing to the pathology of the WOREE syndrome. The value of this report would increase substantially if the authors could provide evidence that at least some of the mutations associated with human disease cause WWOX deficiency or at least discuss how these mutations might affect protein stability based on structural modeling data.
Response to comment #2: Thank you for important comments to increase the value of our manuscript. Since we have focused on analyzing original mutant rat strain, we have no original genetic data directly obtained from human diseases. Although genetics and phenotype of human disease related with Wwox mutation have been summarized in several papers, the relationship between type of mutation and phenotype is still unclear. We have suggested that aberrant C-terminal amino acid sequence resulted from 13-bp deletion in exon 9 might cause degradation of the protein by ubiquitin proteasome system, which is major pathway for aberrant protein degradation, since no Wwox protein has been detected in lde/lderats in previous report. The similar type of mutation has not been reported in human. In this report, we found faint signal consisting with that of mutated protein in size, but the expression level was extremely low. Considering phenotypic similarity of lde/lderats to those of human congenital diseases and knockout mouse, we came to conclusion that phenotypes of lde/lderats is resulted from loss-of-function mutation of Wwox. We are also interested in the degradation process of mutant Wwox protein. But this is not major subject of the present report. We focused on cortical pathogenesis caused by loss-of-function of Wwox. We would like to address the degradation process of the mutated protein in future study.
Comment #3: In addition, the author’s attention is drawn to the following points: 1. Fig 2 and discussion lines 225-237: The authors suggest that the 13bp deletion in exon 9 leads to the translation of a larger and unstable protein product. If the slower migrating band in Ide/Idebrain lysates is due to the expression of a larger mutant protein that would differ from wt by ~1kDa, this band should also be visible in the Ide/+ lysates where currently it might be obscured by the faster migrating band. The authors should attempt to resolve this using gradient SDS-PAGE gels and should quantify the relative amounts of wt and mutant WWOX in all samples. The readers would also benefit from seeing the lower part of the gel to determine if any cleavage products, indicative of protein instability, can be detected in lysates obtained from Ide/+and Ide/Iderats. The should also attempt to convincingly demonstrate the mutant protein instability. This can easily be done for ex. By incubating dissociated liver cells with and without proteasome and lysosome inhibitors for increasing amounts of time and following the accumulation of the slower migrating WWOX band.
Response to comment #3: Thank you for important comments to clarify the situation of larger band present in electrophoresis. In discussion section, we described that larger size protein may be mutated Wwox protein (lines 242-245). This is our speculation based on electrophoresis image in Figure 2 A and B. This is supported by the immunostaining of lde/ldebrain section with markedly less positive signals (figure 2C). Since the expression level of the protein was markedly low, we had to extend the exposure time extremely to detect the faint signal. Therefore, it was difficult to detect the band in +/ldesamples due to its’ lower level of expression (theoretically half in that oflde/lde) and preferential binding of antibody to normal protein. Furthermore, probably due to methodological limitation, we could not detect any degraded protein in western blot of lde/ldeand +/ldesamples. But it is true that functionally normal Wwox protein is almost absent in lde/lderats. Therefore, loss-of-function of Wwox is the cause of the phenotype. Degradation process of mutant protein is out of the major aim in the present study. However, the experiment suggested by reviewer is very interesting, because liver also express Wwox protein. To conduct this experiment, however, we need more time for cloning the mutant cDNA and expressing the protein labeled for detection in liver cells. We also have concern whether the exactly same degradation process might occur in both brain and liver. So, we would like to make it next subject for our study. In this situation, if it is not permitted to write our consideration about the faint signal in the discussion section, we will delete that sentence.
Comment #4: Fig. 3: The fluorescence data on the figure suggest that, although the toatal number of NeuN+ neurons is comparable, there might be a difference in layer organization in Ide/Ide rats compared to wt and that by PND21 there is a difference in cortical thickness that might be related to changes in layer V. Quantification of NeuN+ cells in each layer and of cortical thickness is necessay.
Response to comment #4: Thank you for important comments to consider effects of Wwox to neuron carefully. According to reviewer’s comment, we measured the thickness of cortex and added the data (bar-graph) in Figure 3F of revised MS. But there was no significant deference in cortical thickness between +/+ and lde/lderats during 5-21 days of age (Line 128-130 in revised MS). So far there is no report showing age-related expression of Wwox and influence of loss of Wwox in cerebral cortex. The purpose of this report is examining total effects of Wwox in cerebral cortex development not specific effects to cortical layer. On the other hand, young adult lde/lderats unexpectedly survived long time show reduced thickness of cortex. We speculate the progression of atrophy occurring after 21 days. We would like to remain these subjects to future study, after preparing methodology of precise classification of layers in cortex.
Comment #5: Fig. 5: Iba 1 staining failed and should be optimized and re-evaluated before drawing conclusions. The authors need to provide quantification of GFAP and Iba1+ cell densities, not just a quantification of the fluorescent area.
Response to comment #5: This quantification was done basically based reference #50 (reference #52 in revised MS). In this reference, the areas of astrocytes and microglia were measured and compared between Wwox-knockout and normal mice. The area is influenced by number of cells and size of each cells. In glioma of these cells, not only cell number increased but also cell size increased by activation. That is the reason why we used area-quantification methods as with the reference. We did want to highlight the difference of our results with that in knockout mouse. But we did not get any confidence evidence suggesting glial cell activation. So, in revised MS, we quantified the numbers of GFAP-positive astrocytes and Iba1-poitive microglia in cerebral cortex as according to reviewer’s suggestion. Therefore, we had to use photograph with nuclear counter stain for the quantification of cell number this time. So, we replaced both GFAP and Iba1 photographs to higher magnification with DAPI staining. Although it might be still difficult to recognize cell morphology of each cell due to low magnification, we believe that readers can confirm the quality of our immunostaining in figure 6.
Comment #6: The introduction should provide more information regarding the biological functions of WWOX (e.g. role in signaling, functions of nuclear WWOX, etc).
Response to comment #6: Thank you for your comment to improve introduction. According to reviewer’s comment, we added short sentence for molecular function of Wwox. However, this study focused on representing in vivo data related with loss of Wwox protein in brain but not directly addressing molecular mechanism related to cell signaling and nuclear function in cell culture. Therefore, we added only short sentence of molecular function in Wwox (Line 36-38).
Comment #7: Results, lines 91-92: The data presented in this section are insufficient to speculate that Wwox protein is important for postnatal CNS development. This conclusion can only be reached later.
Response to comment #7: Before the sentence, it was shown that Wwox protein was expressed in most portions of brain, and that the level was gradually increased during early postnatal period. On the other hand, later experiment focused on cerebral cortex not whole brain. This is the reason why we described as like that. Therefore, we would like to remain this sentence with minor change as followed “These findings suggest that Wwox protein might has important roles in normal brain development during early postnatal period (Line 93-95).”
Comment #8: Fig. 1 A: change “Adrenal grand” to “Adrenal gland”.
Response to comment #8: Thank you for finding mistype. It has been corrected in revised MS.
Comment #9: Fig. 1 D: the label for cortical layer I should be moved up.
Response to comment #9: Thank you. It is our mistake. It has been corrected in revised MS.
Comment #10: Line 124: repace “cortexes” with “cortices”
Response to comment #10: Thank you for finding the mistake. It has been corrected in revised MS (Line 132).
Comment #11: Line 151: “around the WM” is vague. If corpus callosum was evaluated, please state so. If a more extended area was quantified, it should be boxed on the figure.
Response to comment #11: Thank you for finding vague expression. Actually, the area used in cell count did not include corpus callosum. The area was cortex including WM. This was mentioned in Material and Methods (Line 368-369 in revised MS, …counted using Image J in same size of brain area shown as representative pictures in each figure). To describe more clearly, we added the sentence defining the region used in quantify (Line 159-161). Actually, we found similar reduction in APC-positive cell number of corpus callosum. We described this finding in revised MS as data not shown (Line 161-162).
Other: We deleted several sentences in the Materials and Methods. That had been remained during revision process in initial version of MS (Line 357-361).
Round 2
Reviewer 2 Report
The manuscript is partially improved still, the authors should make several changes:
As suggested before, even if not addressed experimentally, the effect of WOREE mutations on protein structure and function should at least be discussed, e.g. where the mutations are located and how they could affect function.
Since the quantification if Fig 3 F indicates no change in cortical thickness at P21, the corresponding panel in 3A (bottom right) should be replaced with a more representative image.The statement " The purpose of this report is examining total effects of Wwox in cerebral cortex development not specific effects to cortical layer." is difficult to accept if any real interest is given to the role of Wwox in cortical development. Cortical layers can be distinguished easily based on their morphology and the density of NeuN positive cells in each layer should be determined.
Author Response
Response to the comments from reviewer 2 on manuscript ijms-541172
Comment #1: The manuscript is partially improved still, the authors should make several changes.
Response to comment #1:Thank you for careful reading and important comments. Our responses to Reviewer 2 are described below.
Comment #2: As suggested before, even if not addressed experimentally, the effect of WOREE mutations on protein structure and function should at least be discussed, e.g. where the mutations are located and how they could affect function.
Response to comment #2: Thank you for important comments. We carefully read papers regarding with WOREE syndrome and well considered about the comments again. We added several sentences of discussion based on genetic information of WOREE and changed the discussion of LDE mutation (line 244-259). Actually, functionally null mutations cause WOREE phenotype, even though many type of mutation has been reported in human. Based on the situation, we concluded that lde/lderat is a model for WOREE syndrome.
Comment #3: Since the quantification if Fig 3 F indicates no change in cortical thickness at P21, the corresponding panel in 3A (bottom right) should be replaced with a more representative image.
Response to comment #3: According to the reviewer's comment, we replaced the photograph (Fig 3A, PND21) with more representative one.
Comment #4: The statement " The purpose of this report is examining total effects of Wwox in cerebral cortex development not specific effects to cortical layer." is difficult to accept if any real interest is given to the role of Wwox in cortical development. Cortical layers can be distinguished easily based on their morphology and the density of NeuN positive cells in each layer should be determined.
Response to comment #4: According to the reviewer's comment, we counted neuron number in each cortical layer and showed the bar-graph in Fig.3G of revised manuscript. There is no significant difference between normal and lde/ldein neuron number of each cortical layer. We added several sentences and words related with this experiment in Result, Figure legends, Discussion, and Methods.